# Stochastic Modelling Frameworks for Dragon Fruit Supply Chains in Vietnam under Uncertain Factors

Tri-Dung Nguyen [1,2], Uday Venkatadri [1], Tri Nguyen-Quang [2,*], Claver Diallo [1], Duc-Huy Pham [3], Huu-Thanh Phan [4], Le-Khai Pham [4], Phu-Cuong Nguyen [5] and Michelle Adams [6]

1  Department of Industrial Engineering, Dalhousie University, Halifax, NS B3H 4R2, Canada; tdnguyen@dal.ca (T.-D.N.)
2  Biofluids and Biosystems Modeling Laboratory (BBML), Department of Engineering, Faculty of Agriculture, Dalhousie University, Truro, NS B2N 5E3, Canada
3  Edward P. Fitts Department of Industrial and Systems Engineering, North Carolina State University, Raleigh, NC 27695-7906, USA
4  Postgraduate Studies Office, Ho Chi Minh City University of Technology (VNU-HCMC), Ho Chi Minh City 740030, Vietnam
5  Faculty of Computer Science, Dalhousie University, Halifax, NS B3H 1W5, Canada
6  School for Resource and Environmental Studies, Dalhousie University, Halifax, NS B3H 4R2, Canada
*  Correspondence: tri.nguyen-quang@dal.ca

**Abstract:** Managing uncertainties and risks is always a difficult but fascinating task in fresh fruit supply chains, especially when dealing with the strategy for the production and conveyance of fresh fruit in Vietnam. Following the COVID-19 outbreak, the confluence of economic recession and persistent adverse weather conditions has exacerbated challenges faced by dragon fruit cultivators. This research investigates a two-stage stochastic programming (TSSP) approach which is developed and served as a valuable tool for analyzing uncertainties, optimizing operations, and managing risks in the fresh fruit industry, ultimately contributing to the sustainability and resilience of supply chains in the agricultural sector. A prototype is provided to illustrate the complex and dynamic nature of dragon fruit cultivation and consumption in Vietnam. Data on the selling prices of dragon fruit were collected from several sources between 2013 and 2022 in Binh Thuan Province, Vietnam. The results were obtained from the model by using three different approaches in order of their versatility and efficacy: (1) Scenario tree generation; (2) Sample average approximation; (3) Chance-constrained programming.

**Keywords:** dragon fruit (DF); two-stage stochastic problem; scenario tree generation; sample average approximation; chance-constrained programming

## 1. Introduction

Effectively managing a supply chain that can be impacted by volatile circumstances poses a significant challenge for all involved stakeholders. The supply chain of agricultural products, particularly fresh agricultural products, is inherently challenging due to its susceptibility to numerous unknown and unpredictable elements, such as climate change influences. A farm's longevity and production are adversely impacted by various factors, including drought, seawater intrusion, dangerous pests, illnesses, and overutilization of fertilizers and pesticides.

Uncertainty can manifest at several stages within the fresh fruit supply chain, encompassing production by farmers, post-harvest storage, processing, transportation, and distribution. Furthermore, it is worth noting that the level of uncertainty varies at each point and stage of the supply chain.

The year 2020 presented a multitude of arduous and demanding circumstances within the realm of agriculture, which have been unprecedented in nature for all involved individuals. The global outbreak of the coronavirus pandemic had a significant impact on

farmers across several countries, while distinct meteorological conditions posed challenges to individual locations. The movement of agricultural products from production sites to end consumers shows significant challenges due to constraints on mobility, disruptions in supply chains, restrictions on borders or ports, escalating transportation expenses, and the closure of numerous markets.

The distribution of fruit and vegetable production is subject to significant fluctuations in both demand and prices. According to a report by the FAO in 2020, there has been a significant increase in the pricing of certain items, particularly those that are seen to have immune-boosting properties such as garlic, ginger, and all fruits rich in vitamin C. Conversely, the prices of other products have experienced a sharp decline [1].

The implementation of travel restrictions or limitations between nations exacerbates labor shortages, particularly in countries that heavily depend on seasonal labor [2]. Insufficient labor resources for timely harvesting may result in the spoilage of produce within the agricultural field. Delays in transit and unloading might result in damage to fresh produce while it is being stored in containers.

According to [3], there appears to be a shift in consumer purchasing patterns as a result of imposed limitations on travel. During the initial stages of the pandemic, there was a significant surge in consumer demand for stockpiling commodities, driven by dread and apprehension. Consequently, the fruit and vegetable market experienced a decline, with prices beginning to decrease. This can be attributed to customers purchasing larger quantities of merchandise, thereby contributing to the weakening of the market. Perishable fruits and vegetables saw reduced consumer demand, while their non-perishable counterparts, such as apples and carrots, exhibited greater purchasing power and were prone to price escalation.

### 1.1. Context of the Vietnamese Agriculture

The emergence of the COVID-19 epidemic exacerbated the existing challenges faced by Vietnam's agriculture sector, mostly manifested in decreased output levels and disruptions in agricultural supply chains. In rural regions, there has been a notable increase in the supply of agricultural commodities, including vegetables, flowers, fruits, and seafood, as a consequence of diminished consumer demand. Consequently, these excess commodities remain unutilized and, in certain instances, are subject to destruction. The oversupply has had a discernible influence on the market, resulting in a substantial decrease in prices, particularly for perishable agricultural commodities such as vegetables, flowers, fruits, and seafood. The disparity in pricing between farmers' selling prices and consumers' purchasing prices can be related to issues experienced within the circulation and distribution sector. The escalation of rice prices in the international market can be attributed to the surge in import demand from various nations. Consequently, this surge has resulted in a parallel increase in local prices, particularly for rice [4].

The data provided by The Ministry of Industry and Trade of Vietnam [5] shows that, in 2020, the export value of significant agricultural and fishery products had a decline compared to 2019. Specifically, the seafood industry generated a total revenue of USD 8.41 billion, experiencing a decline of 1.5%. Fruits and vegetables, on the other hand, reached a revenue of USD 3.27 billion, showing a significant decrease of 12.7%. Cashew nuts achieved a volume of 515 thousand tons, resulting in a turnover of USD 3.21 billion, which increased by 13.0% in volume but decreased by 2.3% in turnover. Coffee production amounted to 1.57 million tons, with a turnover of USD 2.74 billion, representing a decline of 5.6% in volume and 4.2% in turnover. Pepper production reached 285 thousand tons, generating a turnover of USD 661 million, which increased by 0.4% in volume but decreased by 7.5% in turnover. Lastly, tea production amounted to 135 thousand tons, with a turnover of USD 218 million, experiencing a decline of 1.8% in volume and 7.8% in turnover.

In the context of Vietnam, a nation mostly reliant on agriculture, it is noteworthy that the primary revenue stream for numerous farmers has also confronted adverse effects. Since the commencement of 2020 till the present, the agricultural industry has experienced

significant disruptions to its production and commercial operations, thereby impacting the financial well-being and livelihoods of farmers. The occurrence of disease poses significant challenges to various operations, including the provision of raw materials, trading, transportation, distribution, and exportation of agricultural products. Numerous enterprises, production facilities, and commercial establishments have experienced temporary closures or terminated contractual agreements, resulting in significant adverse effects on agricultural production. There is a significant prevalence of underemployment and unemployment among laborers in agricultural producing businesses, leading to a substantial decline in the average income of workers [6].

### 1.2. The Need for Mathematical Model of Fresh Fruit Supply Chains in the Vietnamese Context

The use of mathematical models provides substantial advantages in improving the production and distribution of fresh fruit. By tackling the obstacles and making well-informed decisions, these models may have a significant impact on establishing a fruit supply chain that is more effective, environmentally friendly, and adaptable. Nguyen et al. [7] stated that a wide range of mathematical models have been used in the last four decades to identify the most effective solutions for different requirements within the fresh fruit supply chain. Previous review papers [8–12] on mathematical models applied to the agri-food supply chain demonstrate that stochastic models are capable of efficiently addressing challenges including risk and uncertainty.

This paper, in consideration of all above-mentioned contexts, aims to examine the uncertainties associated with the production and distribution of fresh fruit in Vietnam. It specifically focuses on the cultivation of dragon fruit as a case study and the use of a two-state stochastic programming model was involved to respond to the need for sustainable solutions under uncertainties. The deterministic model developed by Nguyen et al. [13] served as a valuable tool for analyzing uncertainties, optimizing operations, and managing risks in fresh fruit production and distribution. Ultimately, this model has contributed to the sustainability and resilience of supply chains in the agricultural sector. The results were obtained from the model by using three different approaches in order of their versatility and efficacy: (1) Scenario tree generation; (2) Sample average approximation; (3) Chance-constrained programming. The application of these methods to the two-stage stochastic model is a potential new direction to address the uncertainties that affect the production and distribution of fresh fruit. Based on the derived solutions, a comparison will be made to assist managers or decision-makers in determining the most suitable strategy for their needs.

But, before presenting our conceptual framework and model, we try to describe a panoramic picture of dragon fruit in the Vietnamese market, to review some contexts of mathematical models existing in the literature for this issue.

### 2. Context of Dragon Fruit in Vietnam: Emerging Trends of Modelling Frameworks and Our Conception

#### 2.1. The Impact of Uncertain Factors on Vietnam's Dragon Fruit Industry

*Price factor:* Due to China's status as the primary market for Vietnamese agricultural products, the onset of the COVID-19 epidemic in China in January posed significant challenges for the procurement and exportation of agricultural goods, including dragon fruits, to the border. Shipments destined for the border are currently experiencing congestion due to insufficient customs processing procedures, compounded by the suspension of sea exports. Prior to the commencement of the Lunar New Year, the prevailing market rate for white dragon fruit exceeded USD 1 per kilogram. However, subsequent to the Lunar New Year, the price experienced a significant decrease, plummeting to a mere USD 0.1 per kilogram. Consequently, no traders have shown interest in procuring the aforementioned commodity [14].

Furthermore, the domestic selling price of dragon fruit has a paradoxical nature. An illustrative instance can be observed in the case of dragon fruit with yellow skin, which

represents one of the three primary varieties of dragon fruit available in the Vietnamese market. In 2004, the Southern Horticultural Research Institute (SOFRI), Vietnam conducted an experiment including the importation and cultivation of Hylocereus costaricensis, a kind of dragon fruit characterized by its yellow peel and white flesh. However, the outcomes did not align with the anticipated expectations. The growth of the plants was suboptimal, characterized by thin and slender branches, as well as the production of little fruits weighing less than 200 g of each. Furthermore, the yield was found to be low, with an average of 2–3 kg per pole every season for 3-year-old plants. In order to satisfy the significant domestic demand and cater to the curiosity of consumers, the fruit is imported from Malaysia and retailed at a price that is up to 20 times greater than that of the red skin and white meat variants [15].

Given the perceived market potential of this novel fruit, numerous horticulturists engage in self-propagation and undertake planting endeavors, commencing in 2018 [16]. Nevertheless, because of the lack of prior knowledge in tree care, the trees exhibit a phenomenon wherein they produce flowers but fail to yield any fruit. Alternatively, there exist fruits that possess unattractive physical characteristics, rendering them unsuitable for commercial transactions. According to a dragon fruit garden owner, there was a significant demand for this fruit in the past due to its high price. As a result, farmers expressed interest in cultivating it. However, the unsuitability of the climate, soil, and cultivation practices for this variety, as opposed to the red skin dragon fruit, hindered their ability to achieve the intended outcomes. Growers face the challenge of declining pricing when they possess knowledge of the methods for cultivating yellow-skin dragon fruits, with prices often ranging from equal to or twice that of red meat dragon fruits, provided the fruit exhibits substantial size and an aesthetically pleasing appearance.

Furthermore, the exportation of the Vietnamese yellow-skin dragon fruit remains unfeasible. The underlying cause is that major consumer markets for dragon fruit from Vietnam, such as China, the European Union, and North America, exhibit a lack of preference for this product. China, being the primary consumer of dragon fruit from Vietnam, exhibits a notable preference for red dragon fruits, whether in terms of their skin or flesh.

*Impact of climate change:* Climate change can potentially affect multiple facets of crop water demand, crop growth and production, irrigation water supply, as well as the occurrence of floods, droughts, and heat waves.

The Global Climate Risk Index 2020 indicates Vietnam to be the sixth nation globally in terms of its exposure to climate change and extreme weather events throughout the period spanning from 1999 to 2018. It can be foreseen that climate change will lead to a more frequent occurrence of natural disasters and extreme heat waves in the majority of Vietnam [17].

According to a report by the Asian Development Bank [18], the risks associated with climate change have been found to impact various socioeconomic aspects in Vietnam including water management, pricing, allocation, access to finance, labor cost, and market price.

Binh Thuan Province is situated in the southeastern region of Vietnam, characterized by an extensive coastline and a monsoon-influenced tropical climate, resulting in well-defined wet and dry seasons. The wet season is typically observed between the months of May and October, whilst the dry season spans from November to April. The region has experienced a range of impacts as a result of climate change, including elevated sea levels, heightened temperatures, and alterations in precipitation patterns.

Binh Thuan, being situated in a coastal area, is susceptible to the consequences of escalating sea levels, encompassing coastal erosion, inundation, and the infiltration of saline water into freshwater reservoirs. The available data indicate that there has been a consistent annual increase in the sea level of approximately 3 mm during the period from 1993 to 2008. Projections suggest that, by the year 2050, the sea level is expected to climb within the range of 28 cm to 33 cm [19].

The region, like other parts of the globe, has encountered a rise in average temperatures under the climate change effect. The potential consequences of this phenomenon extend to various aspects, including ecology, agriculture, and human health. The mean temperature had a gradual increase ranging from 0.5 to 0.7 degrees Celsius during the period from 1958 to 2007. It is projected that the average temperature in 2050 will experience a further increase of 0.4 degrees Celsius compared to the average temperature observed in 2020 [19].

The occurrence of soil erosion, desertification, and drought in Binh Thuan can be attributed to a confluence of factors, including rising temperatures, an upsurge in the frequency of sunny days, and intensified hot winds originating from the mainland during the dry season [20].

The average annual temperature data for Phan Thiet, Binh Thuan, as recorded and compiled by Meteoblue [21] from 1979 to 2021 (Figure 1), indicates a discernible trend towards increasing temperatures, with consistently higher values observed since 2010. In addition, the years 1998, 2016, 2019, and 2020 exhibited the highest average temperatures, accompanied by the occurrence of severe and protracted drought conditions [20,22].

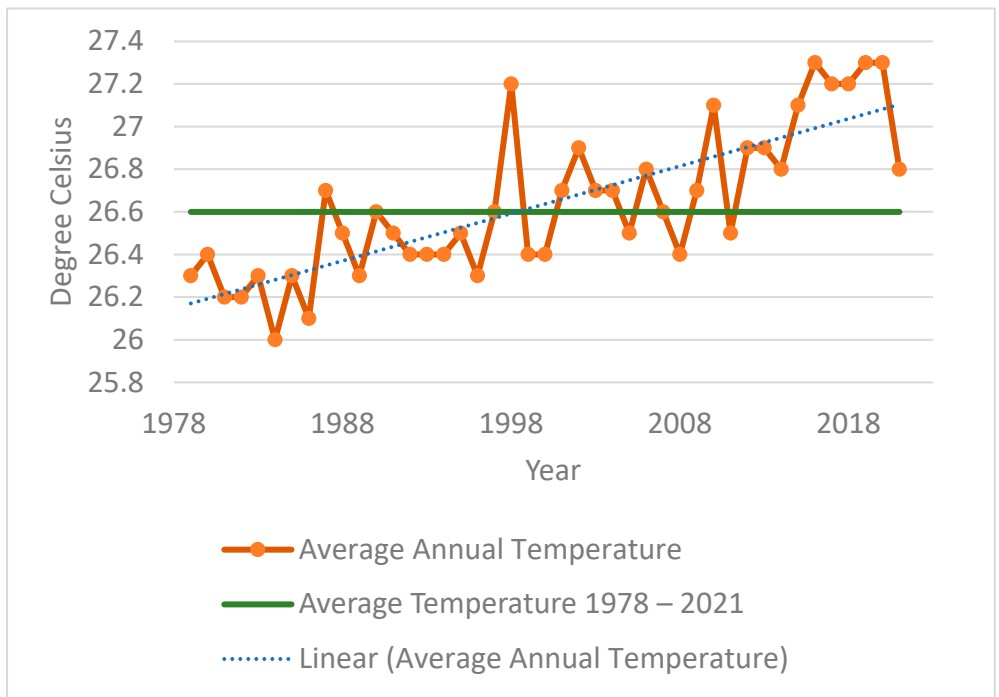

**Figure 1.** Trendline of yearly temperature in Binh Thuan (1979–2021).

The phenomenon of climate change has caused modifications in precipitation patterns, hence giving rise to heightened occurrences of severe weather events including intense rainfall, storms, and periods of drought. Based on the analysis conducted by Doutreloup et al. [23], it is anticipated that the duration of the dry season will be extended due to the projected shift at the beginning and conclusion of the wet season within the time frame of 2046–2065. While there may be alterations in the seasonal patterns of rainfall, it is expected that the overall annual precipitation will remain constant. Similar findings can be drawn for the period spanning from 2081 to 2100. Consequently, the climatic conditions in Binh Thuan Province would undergo alterations characterized by an extended period of aridity, intensified summer precipitation, and heightened occurrences of heavy rainfall.

In recent years, the south–central region has experienced significant rainfall fluctuations attributed to the influence of the El Niño–Southern Oscillation (ENSO) phenomenon. This phenomenon, characterized by the simultaneous occurrence of El Niño and La Niña, has resulted in an increased variability in rainfall patterns. Specifically, the region has witnessed a higher frequency of years with below-average precipitation, leading to a sub-

stantial reduction in total annual rainfall. In some instances, the total annual rainfall has been observed to be more than 20% lower than the long-term average, with certain years experiencing reductions exceeding 30% [24]. In additionally, according to research by Vinh and Huong [25] combined with forecasting models proposed by the Ministry of Natural Resources and Environment of Vietnam [26], drought and water shortage will become more and more serious; specifically, by 2050, there will be no shortage of water. There is still a significant drought area; by 2100, there will be only severe drought in Binh Thuan. Figure 2 below shows a decreasing trend in average annual rainfall between 1979 and 2021 [21].

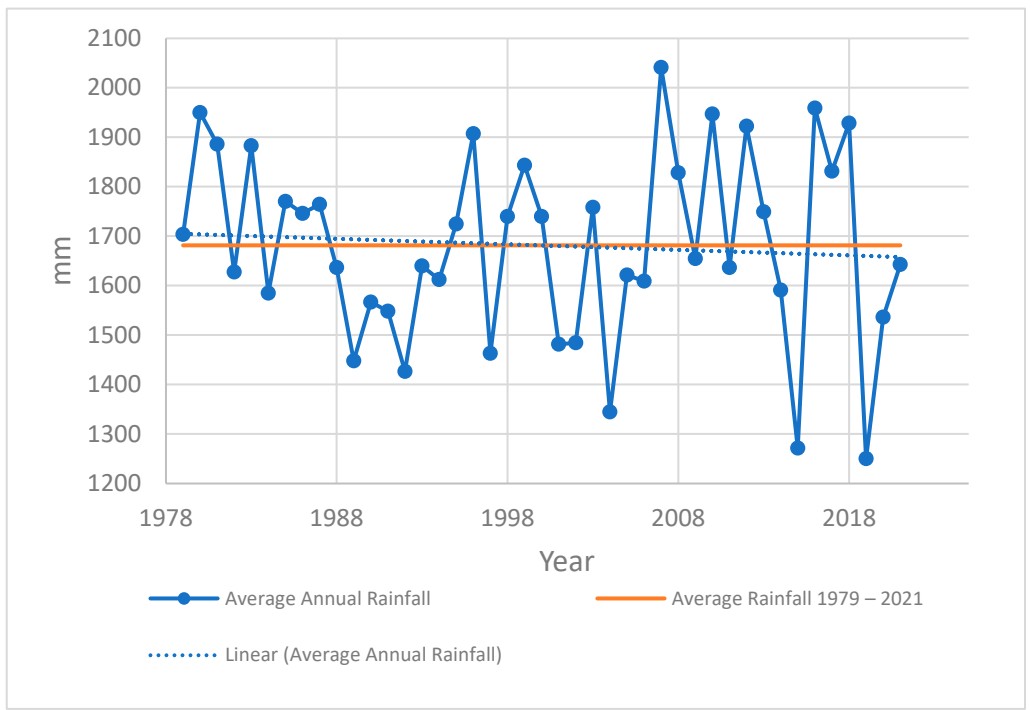

**Figure 2.** Trendline of yearly rainfall in Binh Thuan (1979–2021).

Despite the diminishing duration of the rainy season, the heightened occurrence and magnitude of intense precipitation events can give rise to flash floods and landslides, presenting a significant risk to human lives and assets. Furthermore, an abundance of precipitation has the potential to cause harm to infrastructure and have adverse effects on agricultural productivity. The occurrence of these events can exert substantial impacts on the agricultural sector, water supplies, and infrastructure within the region [27].

The agricultural sector in Binh Thuan Province is susceptible to the impacts of climate change, particularly in relation to crop production. Changes in temperature and precipitation patterns can have detrimental effects on agricultural productivity. Extended periods of drought and severe weather phenomena have the potential to diminish agricultural yields and alter established farming methodologies.

The agricultural sector in Binh Thuan Province is likely to see substantial effects as a result of increasing temperatures. According to the World Economic Forum, the adverse effects of climate change, such as elevated temperatures and intensified precipitation patterns, are causing detrimental impacts on land quality and leading to a decline in soil production. The negative impact on crop production is a consequence of the depletion of organic matter and soil nutrients [28]. Moreover, it is worth noting that alterations in temperature, atmospheric carbon dioxide ($CO_2$) levels, and the occurrence and severity of extreme weather events may exert substantial effects on agricultural productivity [29].

The occurrence of pests and illnesses that have the potential to impact fruit output is being influenced by climate change. For example, alterations in temperature and precipitation patterns have the potential to establish conducive environments for the proliferation

and dissemination of pests and diseases, which can inflict harm upon fruit trees, fruits, and foliage.

In summary, the phenomenon of climate change is exerting various effects on the production of fresh fruit in Binh Thuan. These effects encompass alterations in temperature and precipitation patterns, the incidence of extreme weather events, as well as shifts in the prevalence of pests and illnesses. The aforementioned repercussions have the potential to lead to decreased agricultural production, diminished fruit quality, and financial losses for both farmers and the surrounding community.

### 2.2. Emerging Trends of Modelling Frameworks for Uncertainty and Climate Effects

Managers have been utilizing and developing decision-support systems in response to the uncertainties associated with fresh fruit production/distribution. The persistence of uncertainty and the challenges associated with forecasting continue to be a matter of concern [30].

The agri-food supply chain is a multifaceted system encompassing production, distribution, and consumption. It is marked by considerable volatility arising from various causes, including weather conditions, market fluctuations, and customer preferences. Researchers have additionally presented a diverse range of mathematical models aimed at assisting managers and direct farmers in mitigating errors in decision-making within the context of risks and uncertainties prevalent in the agricultural supply chain. Nguyen et al. [7], who aim to improve the supply chains of agricultural goods, are the main researchers that use the deterministic model. The decisions made by managers, however, may be prone to error due to the inherent nature and limitations of the deterministic model, which lacks the ability to effectively address uncertainties and risks. In recent years, there has been an emergence of stochastic programming and resilient optimization models as viable approaches to address problems characterized by uncertain aspects. The aforementioned methodologies have undergone enhancements, rendering them valuable instruments for decision-makers to promptly and efficiently tackle challenges pertaining to manufacturing, processing, transportation, and distribution.

Stochastic linear programming (SLP) is an extension of linear programming that incorporates parameters with inherent uncertainty. It has been applied in agri-food supply chain management, particularly in crop planning and animal production. Researchers like Carøe and Schultz [31] have developed dual decomposition, a method for breaking down large stochastic integer programs into smaller subproblems. Pourmohammadi et al. [32] created a model considering production, transportation, storage, processing, and regional demand to optimize wheat supply chains under uncertainty. The model reduced costs and improved supply chain performance. Jacquet and Pluvinage [33] developed a discrete stochastic programming model to investigate how climatic variability affects farm management and evaluate farm strategies under different weather conditions. The model optimizes farm income and evaluates agricultural strategies in the setting of climate uncertainties. The study concluded that diversification and insurance policies may help farmers handle climatic uncertainty risks, emphasizing the importance of climate unpredictability in agriculture policy design and decision-making.

Two-stage stochastic programming (TSSP) is a widely used method in supply chain management, particularly in the agri-food industry. It helps manage the trade-off between long-term and short-term decision-making, considering uncertainty. Studies by Darby-Dowman et al. [34], Kazaz [35], Ahumada et al. [36], Tan and Çömden [37], Costa et al. [38], Marchal et al. [39], and Flores and Villalobos [40] have all highlighted the importance of TSSP for managing uncertainty in supply chain management.

Darby-Dowman et al. [34] developed a TSSP model with recourse model for horticulture planting plans, which accounts for weather and crop output variables. Kazaz's production planning model [35] considers yield, demand, cost, and price interdependencies to optimize production decisions. Ahumada et al. [36] optimized production and distribution using a two-stage stochastic mixed-integer linear programming approach (SMILP),

accounting for demand fluctuations, yield variability, and transportation costs. In their work, Tan and Çömden [37] optimized yearly crop planning by addressing multiple sources of uncertainty, including variations in demand, maturation, harvest, and yield hazards. Costa et al. [38] provided a paradigm for managing perishable vegetable crop supply chains, considering agricultural production, transportation, storage, and product perishability. Marchal et al. [39] created a SMILP model to optimize production planning and address uncertainties. Flores and Villalobos [40] suggested a stochastic planning framework to help agricultural stakeholders integrate different systems while considering market prices, crop yields, and weather variables. The approach optimizes land allocation, production, and resource management, maximizing predicted profit while considering land, labor, and environmental constraints.

Robust optimization (RO) is a method that considers multiple scenarios to create less uncertain solutions. Bohle et al. [41], Munhoz and Morabito [42], and An and Ouyang [43] have used RO in agri-food supply chain management to improve resilience and reduce uncertainty-related risks.

A model developed by Bohle et al. [41] optimizes harvest timing to maximize grape quality and minimize labor and equipment expenditures, improving wine grape harvesting schedule efficiency and dependability. Munhoz and Morabito [42] created a robust citrus firm production planning optimization model using recourse actions and two-stage SMILP. The model minimizes the total estimated cost of the production plan, including supply and demand uncertainties, and meets customer demand and product quality standards in uncertain settings. In 2016, An and Ouyang [43] introduced a resilient grain supply chain design model that accounts for post-harvest loss and harvest timing equilibrium, resolving supply–demand uncertainty and harvest timing–crop yield trade-offs. The model accurately represents the complicated harvest timing and post-harvest loss trade-offs, resulting in more robust and efficient supply chain architectures.

Multi-objective Stochastic Programming (MOSP) models have been utilized to manage the agri-food supply chain, focusing on economic, environmental, and social trade-offs. Banasik et al. [44] developed a decision-support tool that considers demand, production yield uncertainty, and environmental factors. They used a mixed-integer linear programming (MILP) methodology to reduce production costs and environmental effects. The model allows producers to adjust their schedule based on demand and yield, resulting in more robust and eco-efficient production plans. Chavez et al. [45] suggested a multi-objective stochastic optimization model for scheduling upstream operations in a sustainable sugarcane chain while considering growth, harvest, transportation, manpower, machinery, and vehicle scheduling. A compromise programming methodology was used to find the best harvesting method trade-offs in a Peruvian case study.

Global warming threatens natural resources, ecosystems, and human society [46]. Climate change is worsening the impact of weather patterns on crop yields, productivity, and quality. Climate change significantly impacts agriculture, including fresh fruit production, due to temperature, rainfall, and extreme weather events [47,48].

Duangdai and Likasiri [49] used mathematical modelling to study the relationship between global temperature and forest coverage, with a significant negative association found between temperature and rainfall. Lim et al. [50] proposed a two-stage optimization model to increase oil palm plantation harvesting and transport efficiency, which reduced journey distance and improved harvesting and transportation efficiency. These findings can help plantation managers allocate resources more efficiently, increasing production and lowering costs. Sun et al. [51] studied climate change and vegetation patterns using mathematical modelling and data analysis, highlighting the need for understanding climate change's complex interactions with vegetation dynamics to develop effective conservation and management strategies. Ghaffari et al. [52] used a Positive Mathematical Planning (PMP) model to evaluate drought's economic consequences on agriculture under various climate change scenarios, emphasizing the need for effective adaptation and mitigation strategies. Kung and Wu [53] examined how climate change influences water allocation

and bioenergy output using stochastic mathematical programming. They found that water availability significantly impacts bioenergy output and that water distribution strategies may have different effects. Climate change management requires efficient water distribution, and adapting to water availability, crop yield, and unpredictability is crucial. This research highlights the importance of managing climate change sustainably to ensure sustainable development of water resources, agricultural productivity, and bioenergy.

As a matter of fact, studies on stochastic programming models for the agri-food supply chain need to include the following:

- Enhanced prediction and decision-making through machine learning and artificial intelligence integration.
- The development of two-stage stochastic programming (TSSP) models to enhance supply chain management.
- The expansion of stochastic models to address climate change, sustainability, and circular economy.

### 2.3. Our Conception and Model

To address the intricacies of decision-making processes and the prevalence of uncertain inputs, several mathematical and statistical methodologies are often explored. Among them, non-linear programming and stochastic programming are particularly prominent. This study introduces and applies stochastic programming and robust optimization techniques, including the scenario tree method, sample average approximation, and chance-constrained formulation. These approaches are utilized to construct a mathematical model that addresses the uncertainty associated with selling prices within the context of the real food system.

The stochastic model developed for a normal growing season can consist of two distinct phases.

The initial stage encompasses strategic choices that are exclusively made at the commencement of the agricultural season, including the selection of crops, determination of optimal planting quantities, and establishment of appropriate cultivation timelines.

During the second phase, farmers make adjustments to the decisions that were previously made in the first phase as the season develops. The farmers are faced with the task of determining the optimal quantity to harvest throughout each season and making decisions regarding which customers to sell their produce to, based on prevailing market conditions. The model additionally incorporates the probabilistic characteristics of the dragon fruit production and distribution processes.

The subsequent phase involves the consideration and analysis of uncertain factors that exhibit either high risks or low probability but large magnitude risks. The variability in risk levels associated with different uncertainties is contingent upon the specific cultivar of dragon fruit, as stated in Table 1.

**Table 1.** Risk level of uncertain factors for each variety of dragon fruit.

| | Variety of Dragon Fruit | | |
|---|---|---|---|
| Factor | Red skin | Red flesh | Yellow skin |
| Weather conditions | Moderate | Moderate | Moderate |
| Yield | Moderate | Moderate | High |
| Market price | Low | Moderate | High |
| Demand | Moderate | Moderate | High |

## 3. Methodology

### 3.1. Conceptual Framework

The proposed models for research on the fresh fruit supply chain in Vietnam are developed based on existing problems of production and processing dragon fruit to deal with the uncertain parameters of the agriculture by using stochastic optimization and robust approaches. The models handle middle- and long-term decisions such as crop planting plans, growing and harvesting plans, and distribution plans for dragon fruit plants in Vietnam. A basic network of dragon fruit production and processing is considered and described in Figure 3 that presents the different combination of transactions between parties of the supply chain network.

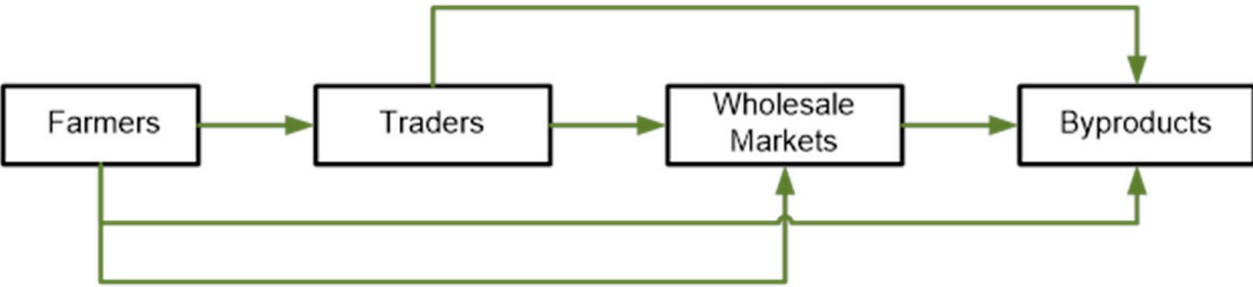

**Figure 3.** Dragon fruit production and processing chain.

Firstly, a recourse stochastic model is extended from the deterministic model proposed by Nguyen et al. [13] and it includes two stages. To make decisions that are reliant on random variables such as the market price, the TSSP approach is suggested and summarized in Figure 4. Due to the characteristics of perennial tropical fruit plants, the first-stage solution for the problem is made considering planting factors such as constraints of land, water, labor, limitations on investment for first years of planting, and annual costs for year-by-year crops. The objective of the first stage is supporting the farmers to evaluate and make better decisions and policies that not only increase the expected revenue but also reduce their risk by considering different scenarios.

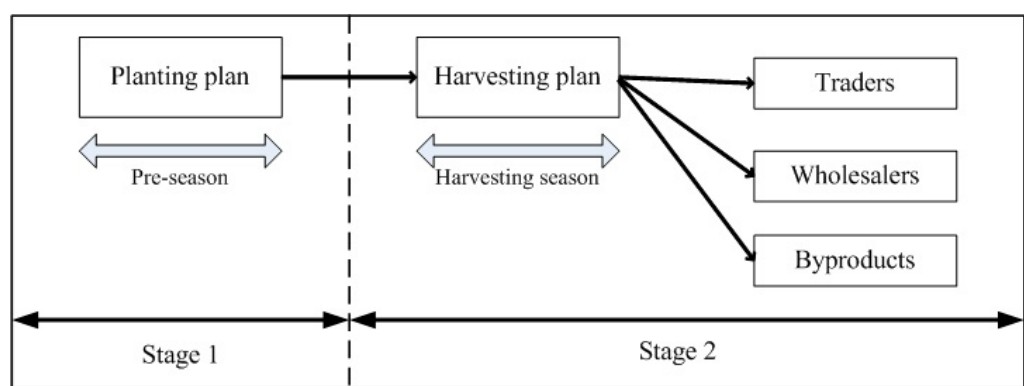

**Figure 4.** The two-stage decision-making process for fresh fruit production and distribution.

In the second stage, to provide satisfactory service to customers, the cognitive process to make decisions requires trustworthy and detailed statistics such as the capacity to harvest and the quantity to ship to selected customers while the prices fluctuate. Then, to better provide detailed distributing plans, reduced production and market uncertainty are approximated using the vicinity of the operational planning to the actual harvesting period. The environment of decision-making of this TSSP model is described in Figure 5.

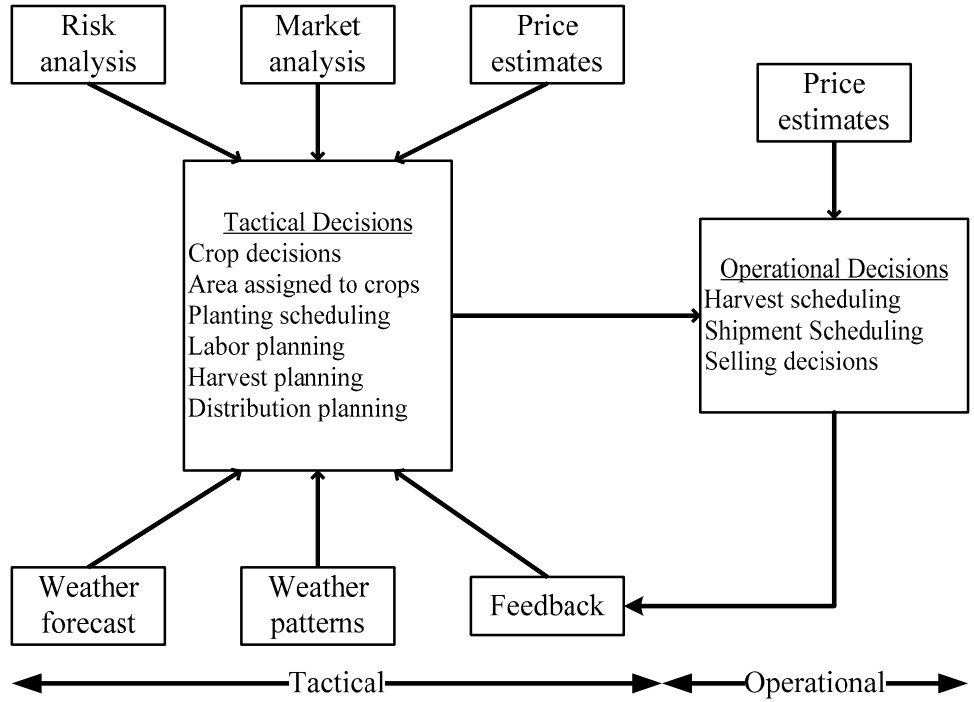

**Figure 5.** The environment of decision-making of the recourse two-stage stochastic model (TSSP).

This study presents two sampling procedures that are proposed as appropriate approaches to tackle the uncertain difficulties associated with fresh fruit production and delivery in Vietnam. The approaches employed in this study include scenario tree generation and sample average approximation. These methodologies are designed to conceptualize and evaluate ambiguity in order to identify feasible resolutions to the issue. The inherent flexibility and scalability of both approaches [54] offer significant advantages, making them effective instruments for addressing difficulties in stochastic programming. Figure 6 shows the methodological steps that are implemented in this paper.

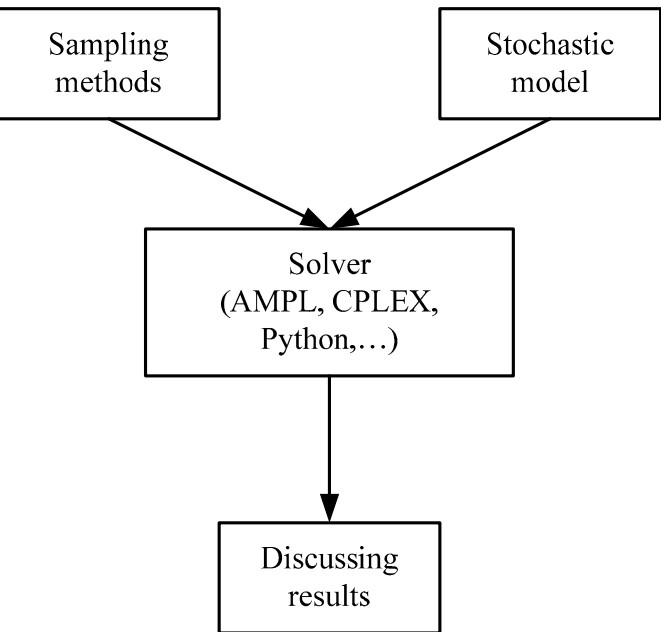

**Figure 6.** Methodological procedures implemented for the study of the production and distribution of dragon fruit in Vietnam.

The model was tested by using a dataset of a Vietnamese fresh fruit to verify the validity of results. The data were collected using methods such as surveys and direct observation. In the modeling phase, the historical data taken from the Department of Agriculture and Rural Development of Binh Thuan Province and General Statistics Office of Vietnam were used to validate the results. To ensure the usefulness and the applicability of the models in the real life, the data used to validate were collected directly from farmers, traders, and wholesalers who are involved in a fresh fruit supply chain by using interviewing or performing surveys.

*3.2. Formulation of the Stochastic Model*

- Objective: maximizing the profit of crop harvesting for T years.

$$
\begin{aligned}
\max \quad & \sum_s \rho_s \Big( \sum_j \sum_i \sum_t p_{jits} ST_{jits} + \sum_j \sum_m \sum_t q_{jmts} SWM_{jmts} + \sum_j \sum_t r_{jts} SB_{jts} \\
& - \sum_j \sum_i \sum_t cPNT1_{jit} \epsilon^1_{jits} - \sum_j \sum_m \sum_t cPNT2_{jmt} \epsilon^1_{jmts} - \sum_t \Big( cbp_t \sum_j SB_{jts} \Big) \Big) \\
& - \sum_j \sum_t cp_{jt} Y_{jt} - \sum_t \Big( ch_t \sum_j \sum_k X_{jkt} \Big) - \sum_t \Big( cr_t \sum_j \sum_k Z_{jkt} \Big) - \sum_t F_t clabf_t \\
& - \sum_t Hire_t clabp_t - \sum_k \Big( cwaterw_k \sum_j \sum_t X_{jkt} \Big) - \sum_j \Big( clightingv_j \sum_k \sum_t X_{jkt} \Big)
\end{aligned}
\tag{1}
$$

The model is designed to optimize decisions around planting and harvesting to maximize predicted revenues for farmers. This highlights the discrepancy in aspects such as revenue projections from selling to markets (SWM), traders (ST), and byproduct providers, compared to the overall expenditures incurred for rooting, truncating, byproduct processing, penalties for demand shortfall, water costs, laboring, and lighting.

- Constraints: the below constraints are applied annually.
  - Land availability.

Each crop has an expanse of $j$ at an age of $k$ that does not exceed the available land ($L$).

$$
\sum_j \sum_k X_{jkt} \leq L
\tag{2}
$$

- Age-class balance in planting

The planning structure of agricultural models is related to the third category of limitations (3)–(8). Catalá [55] modeled the process of chopping down in a study of planting new apple and pear trees.

$Y_{jt}$ resolves the plantation decisions. A freshly cultivated fruit tree is always considered to be of age 0. Constraint (3) specifies that only crops in age class 1 can be planted in year 1.

$$
X_{jkt} = Y_{jt} \quad \forall \begin{cases} 1 \leq j \leq J \\ k = 1 \end{cases}
\tag{3}
$$

Constraint (4) ensures that the newly planted crops are uncut in the same year.

$$
Z_{jkt} = 0 \quad \forall \begin{cases} 1 \leq j \leq J \\ k = 1 \end{cases}
\tag{4}
$$

Only the first year is subject to constraints (5) and (6), while other age classes ($k = 2 \ldots 10$) are not. Based on this, the planted sector is the number of trees in age class $k - 1$ in year

0 less than the amount that can be felled in the first year once the trees have survived for a year.

$$X_{ijt} = I_{j,k-1} - Z_{jkt} \quad \forall \begin{cases} 1 \leq j \leq J \\ 2 \leq k \leq K \end{cases} \tag{5}$$

$$Z_{jkt} = I_{i,k-1} \quad \forall \begin{cases} 1 \leq j \leq J \\ k = K \end{cases} \tag{6}$$

In the planning horizon, periods $t > 1$ are subject to constraints (7) and (8). For crops with age classes $10 \geq k > 1$, constraint (7) is applicable. Relative to the optionally cleared area, the extent of the plantation each year is determined by the preceding year.

$$X_{jkt} = X_{j,k-1,t-1} - Z_{jkt} \quad \forall \begin{cases} 1 \leq j \leq J \\ 2 \leq k \leq K \end{cases} \tag{7}$$

All crops that reach the age of 9 in the considered year $t-1$ should be truncated in the following year, according to constraint (8).

$$Z_{jkt} = X_{j,k-1,t-1} \quad \forall \begin{cases} 1 \leq j \leq J \\ k = K \end{cases} \tag{8}$$

- Constraint (9) makes sure that the total harvest in each scenario is less than the product of the planting area (in hectares) by the yield (metric tons/hectare).

$$\sum_i ST_{jits} + \sum_m SWM_{jmts} + SB_{jts} \leq \sum_j y_{jkt} X_{jkst} \quad \forall \begin{cases} 1 \leq j \leq J \\ 1 \leq s \leq S \end{cases} \tag{9}$$

- In each scenario, constraints (10)–(12) set the demand satisfaction in each year.

$$T_{jits} = djit - \epsilon^1_{jits} \quad \forall \begin{cases} 1 \leq j \leq J \\ 1 \leq i \leq I \\ 1 \leq s \leq S \end{cases} \tag{10}$$

$$SWM_{jmts} = ejmt - \epsilon^2_{jmts} \quad \forall \begin{cases} 1 \leq j \leq J \\ 1 \leq m \leq M \\ 1 \leq s \leq S \end{cases} \tag{11}$$

$$SB_{jts} = f_{jt} \quad \forall \begin{cases} 1 \leq j \leq J \\ 1 \leq s \leq S \end{cases} \tag{12}$$

- Labor constraints

$$F_t + Hire_t - P_t \cdot \sum_j Y_{jt} - H_t \cdot \sum_j X_{jt} - R_t \cdot \sum_j Z_{jkt} = 0 \qquad \forall t \tag{13}$$

Constraint (13) stands for the labor requirements to plant, cut, and harvest in a given year, in which

$$F_t = M \qquad \forall t. \tag{14}$$

There may occasionally be a set number of full-time employees. In this case, it is expressed using

$$Hire_t \leq N \qquad \forall t. \tag{15}$$

The requirements of cultivating, harvesting, or truncating could determine the hiring of the part-time workforce. However, as noted in constraint (15), this number is restricted by the limits of the budget.

- Water limitation

$$\sum_j \sum_k X_{jkst}.w_{jkst} \leq W_{st} \qquad \forall t, s \qquad (16)$$

For each age *k* of a tree, the water requirement per hectare for a specific crop *j* varies in season *s* of year *t*. The variables must not surpass the water available, since excessive watering of trees leads to decreased output.

- Lighting limitation

$$\sum_j X_{js}.v_s \leq V_s \qquad \forall s \qquad (17)$$

*3.3. Scenario Tree Generation*

The method of generating scenario trees is a highly effective tool in the field of stochastic programming, and it has been extensively employed in the resolution of various practical issues. The method in question possesses characteristics that render it adaptable, interpretable, accurate, and scalable.

The utilization of the scenario tree generation technique is prevalent in various domains such as finance, operations research, and strategic planning. This tool facilitates the ability of decision-makers to create models that account for uncertainties, generate estimations of potential outcomes, and enhance the process of decision-making across a range of conditions. The process entails the creation of a hierarchical arrangement characterized by interconnected nodes, wherein each node corresponds to a distinct decision, event, or consequence.

The methodology was initially presented by Howard Raiffa in his publication titled "*Decision Analysis: Introductory Lectures on Choices under Uncertainty*" [56]. The concept has been subsequently extended and improved upon by several academics, such as Dantzig and Infanger [57] who employed the technique to address complex linear programming issues on a significant scale.

In a study, Calfa et al. [58] introduced a novel approach to construct multi-stage scenario trees that maintain consistency with both historical and projected data. The methodology employed in this study follows a two-step process. Firstly, statistical property matching is utilized to develop a collection of scenarios that exhibit statistical features similar to those observed in the historical data. Secondly, a Distribution Matching Problem is addressed to verify that the generated scenarios also align with the forecasted distribution of the data. The method suggested in this study presents a novel and effective approach for generating scenario trees in the context of multi-stage stochastic programming issues. The methodology is based on robust statistical principles and demonstrates computational efficiency. Consequently, it will be selected for implementation in the context of optimizing the production and distribution processes pertaining to fresh fruit in Vietnam.

In this paper, based on the algorithm proposed by Calfa et al. [58], a Distribution Matching Problem (DMP) model is developed using the Moment Matching Problem combined with the Empirical Cumulative Distribution Function (ECDF) and is designed to be applied to build scenario trees based on the fresh fruit price dataset collected in Vietnam. The procedure of the algorithm and an illustrated scenario tree are depicted in Figures 7 and 8 below.

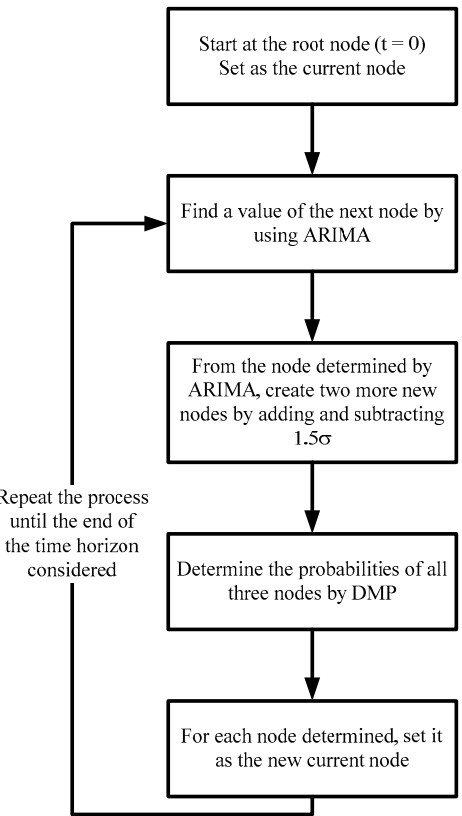

**Figure 7.** The procedure of building a scenario tree for the selling price of the dragon fruits.

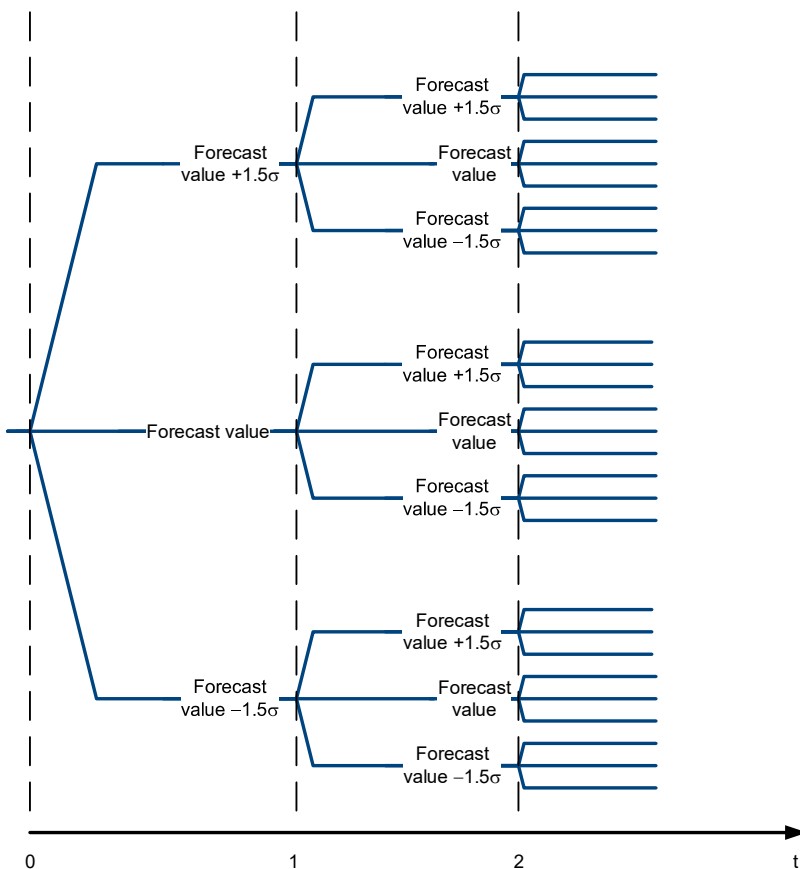

**Figure 8.** Illustrated scenario tree for the selling price of the dragon fruits.

The objective of using the Distribution Matching Problem is to determine the ideal values of random variables and probabilities associated with scenario trees, with the aim of minimizing discrepancies between statistical features derived from the tree and those computed directly from the available data [58]. The mathematical model of the DMP is described in Appendix B.

### 3.4. Sample Average Approximation

In stochastic optimization issues where the objective function cannot be determined precisely but can be estimated through simulation, sample average approximation (SAA) is a common approach. To solve the approximate problem using deterministic optimization techniques, the original problem is replaced by an approximation based on a finite sample of random scenarios. With independent training samples, SAA has strong asymptotic performance guarantees, but these assurances might not be universally true with dependent samples [59,60].

The SAA method in our study uses Monte Carlo simulation to address optimization problems with stochastic elements. In this approach, the predicted objective function is estimated by calculating the sample mean from a random sample [61]. To deal with the fresh fruit production and distribution problem, the SAA operates by producing a collection of scenarios that accurately depict the outcomes of uncertain price fluctuations.

The procedure for applying SAA is described in Figure 9, as follows.

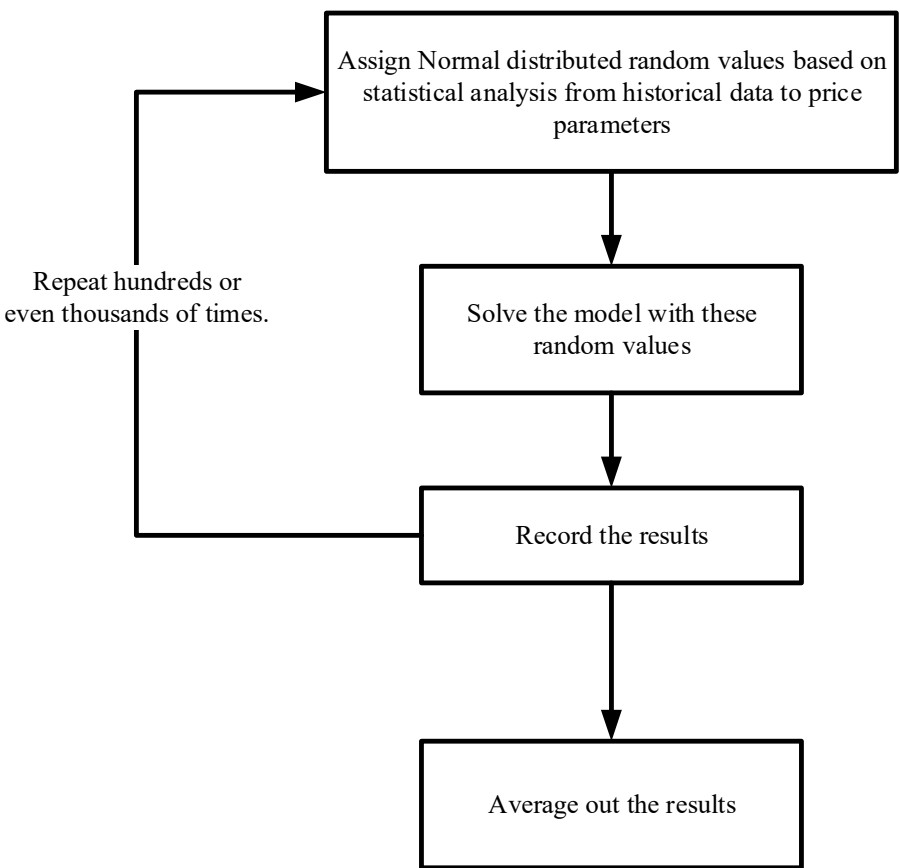

**Figure 9.** The procedure for applying SAA for the production and distribution fresh fruit stochastic model.

### 3.5. Chance-Constrained Programming

Chance-constrained programming (CCP) is a robust and adaptable optimization technique utilized to formulate and address stochastic programming issues in the presence of uncertainty [62]. The optimization framework is enhanced by the inclusion of probabilistic

constraints, which guarantee that the solution adheres to a specified probability threshold for constraint satisfaction [63].

The idea of CCP was initially mentioned by Markowitz [63] to address portfolio optimization issues. In their seminal paper on optimizing oil distribution systems, Charnes and Cooper [64] formally established CCP, marking a significant milestone in the field of stochastic optimization. Subsequently, this concept has been expanded to encompass a wide range of applications, such as energy management, supply chain planning, financial risk management, and project scheduling.

The issue of fulfilling demand was resolved with the implementation of a penalty system for any instances of insufficient supply. Our objective is to ensure that the requirements of every product from each dealer are met with a probability that exceeds a specified level of reliability (*Rel*). The concept is denoted as a chance constraint.

Let Rel be a desirable dependability value, where $0 \leq Rel \leq 1$. The presence of shortage factor $\varepsilon$ is no longer necessary; yet, it remains crucial to ascertain whether the demand will be satisfied or not. In the context of the chance-constrained deterministic formulation, it is necessary to introduce binary variables to accurately describe the chance constraint. In this section, therefore, we redefine $\epsilon$ as a binary variable.

- $\epsilon_{jits}^{1}$ is 1 if a shortage of crop $j$ happens at trader $i$ in period $t$ for scenario $s$—0 otherwise.
- $\epsilon_{jmts}^{2}$ is 1 if a shortage of crop $j$ happens at wholesaler $m$ in period $t$ for scenario $s$—0 otherwise.

Since the deficiency penalty no longer needs to be minimized, the new objective function (18) developed and modified from the objective function (1) is introduced as follows:

$$
\begin{aligned}
\max \sum_{s} \rho_{s} (&\sum_{j}\sum_{i}\sum_{t} p_{jits} ST_{jits} + \sum_{j}\sum_{m}\sum_{t} q_{jmts} SWM_{jmts} + \sum_{j}\sum_{t} r_{jts} SB_{jts} \\
&- \sum_{t}\left(cbp_{t}\sum_{j} SB_{jts}\right)) \\
&- \sum_{j}\sum_{t} cp_{jt} Y_{jt} - \sum_{t}\left(ch_{t}\sum_{j}\sum_{k} X_{jkt}\right) - \sum_{t}\left(cr_{t}\sum_{j}\sum_{k} Z_{jkt}\right) - \sum_{t} F_{t} clabf_{t} \\
&- \sum_{t} Hire_{t} clabp_{t} - \sum_{k}\left(cwaterw_{k}\sum_{j}\sum_{t} X_{jkt}\right) - \sum_{j}\left(clightingv_{j}\sum_{k}\sum_{t} X_{jkt}\right)
\end{aligned}
\tag{18}
$$

Furthermore, to ensure the attainment of the necessary level of reliability, the following constrains are incorporated:

$$
\sum_{s} \rho_{s}.\epsilon_{jits}^{1} \leq 1 - Rel^{1} \quad \forall \begin{cases} 1 \leq j \leq J \\ 1 \leq i \leq I \\ 1 \leq s \leq S \end{cases} ;
\tag{19}
$$

$$
\sum_{s} \rho_{s}.\epsilon_{jmts}^{2} \leq 1 - Rel^{2} \quad \forall \begin{cases} 1 \leq j \leq J \\ 1 \leq m \leq M \\ 1 \leq s \leq S \end{cases} .
\tag{20}
$$

However, a limitation of the above constraints is the absence of a guarantee that variables $\epsilon$ will assume a value of 1 in the event of a shortage. Constraints (10) and (11) will be rewritten to satisfy the requirement that, when there is a shortage, the variable $\epsilon$ will have the value 1:

$$
d_{jit} - ST_{jits} = 10,000 \epsilon_{jits}^{1} \quad \forall \begin{cases} 1 \leq j \leq J \\ 1 \leq i \leq I \\ 1 \leq s \leq S \end{cases} ,
\tag{21}
$$

$$e_{jmt} - SWM_{jmts} = 10,000\epsilon_{jmts}^2 \quad \forall \begin{cases} 1 \leq j \leq J \\ 1 \leq m \leq M \\ 1 \leq s \leq S \end{cases}. \tag{22}$$

Constraints (2)–(9) and (12) of the stochastic model remain unchanged.

Finally, the rewritten model with probabilistic constraints is solved by using the sample average approximation method with 200 iterations.

The case study aims to optimize the production and distribution of dragon fruit from growers to traders, wholesalers, and byproducts, with the objective of maximizing profits. To achieve this, three approaches are employed: scenario tree generation, sample average approximation, and chance-constrained programming. These methodologies are utilized to identify potential solutions for the case study. The obtained results from the used methodologies are presented and analyzed to provide a comprehensive conversation. This discussion aims to offer help and guidance to growers or managers involved in the production of dragon fruit, aiding them in making informed decisions.

The stochastic model was coded and solved in Python language on a computer configured with a 12th Gen Intel (R) Core (TM) i7 Processor and RAM of 32.0 GB.

## 4. Results and Discussion

The results herein obtained are, in fact, the extension study of the scenarios deterministically solved and presented in Nguyen et al. [13]. These findings can assist farmers in making decisions regarding the allocation of land for several varieties of dragon fruit throughout a 20-hectare area over an 8-year period with fluctuating selling prices. As mentioned above, selling prices of dragon fruits completely plunged during the outbreak of the COVID-19 pandemic; the question " how dragon fruit growers would act if a similar thing were to happen, and how they could maximize their profit as well as minimize their risks in the future", requires further investigation.

According to Nguyen et al. [13], three kinds of dragon fruit are planted for local demands and exporting including red-skin, white-flesh (Crop 1); red-skin, red-flesh (Crop 2); and yellow-skin, white-flesh (Crop 3). Since China is the largest dragon fruit importer market, the prices are dominated mostly by Chinese traders. The selling price agreed between the growers and the traders is mainly based on trust (verbal contract); if the trader terminates the contract because of finding a better source of dragon fruits, the farmer will suffer.

Before the COVID-19 pandemic in January 2020, the price of Crop 1 remained stationary during episode 1 (peak season) and slightly rose in episode 2 (off-peak season). Crop 2 was priced double as high due to its cultivation for sale to the Chinese markets, where there is an extremely high demand. The price of Crop 3 cultivated in Vietnam was below expectations. Vietnamese farmers then aimed to sell their locally grown fruits at a price equivalent to the selling price of Crop 3 imported from Malaysia, which is 20 times greater than white-flesh dragon fruits. Its greatest price is equivalent to the price of red-flesh ones.

During the pandemic period, the prices of all three kinds of dragon fruit dramatically fell when the lockdowns were declared due to the outbreaks. Many farms did not have traders, so ripe dragon fruits were used as food for cattle or on a big sale— USD 0.1–0.2 per kilogram. At that selling price, the growers do not have enough revenue to compensate for production costs for the next season.

The current study presumes that Crop 1 is the established supply chain for both local and international trade with a recognized measurable demand. Crop 2 is exclusively cultivated for importation into Chinese markets, whereas Crop 3 is grown to assess its market potential. The selling prices of all crops in the model were assumed to randomly vary following a normal distribution curve over the years.

Establishing a specific land ratio for different types of dragon fruit is envisaged to help dragon fruit growers protect their profits from market price fluctuations caused by such COVID-19 pandemic events or severe unpredicted weather conditions resulting from climate change in the next several years.

### 4.1. Scenario Tree Generation

A look at historical data shows the variation in selling prices of red-skinned, white-fleshed dragon fruit and red-fleshed, red-skin dragon fruit, with monthly average data from July 2019 to December 2022 (Figure 10) and average annual data from 2013 to 2022 (Figure 11).

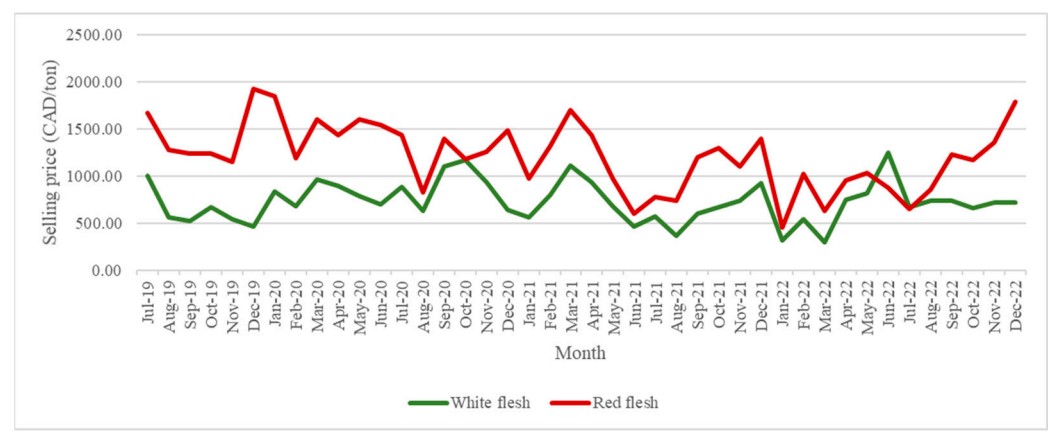

**Figure 10.** Historic data of monthly selling prices of white-flesh, red-skin and red-flesh, red-skin DF.

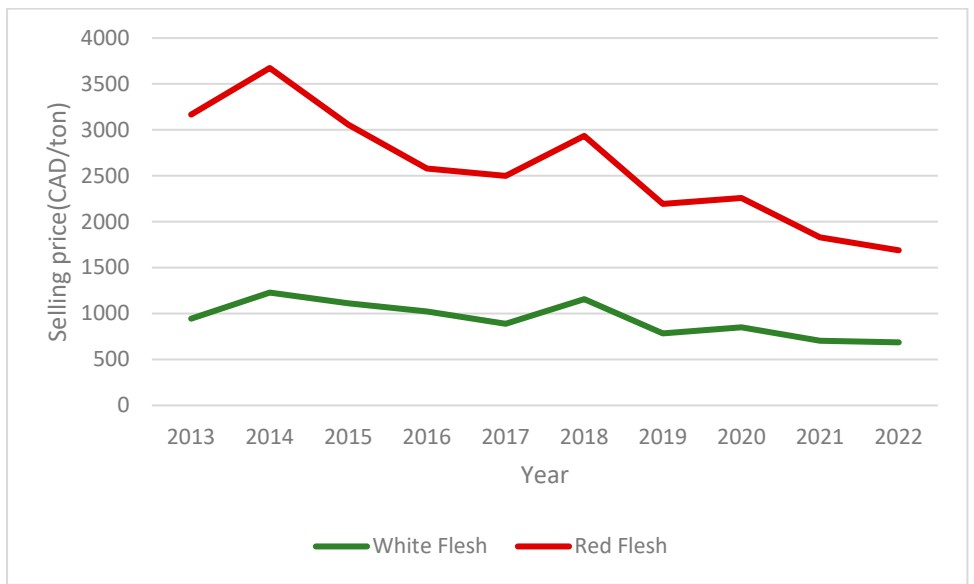

**Figure 11.** Historic data of yearly selling prices of white-flesh, red-skin and red-flesh, red-skin DF.

An uncomplicated method for constructing the framework of a multi-period scenario tree is to have three potential outcomes at each node. These outcomes consist of the predicted value as well as two values obtained by adding and subtracting 1.5 standard deviation from the predicted value for each period. The predicted value is derived from the ARIMA model, with a 95% level of confidence and the parameters p, d, and q set to 1, 0, and 0, respectively. Since the selling price data are applied based on the ARIMA model, it is expected that the forecasted value will conform to a normal distribution. The standard deviation ($\sigma$) value is derived from statistical analysis of historical data on dragon fruit selling prices in Binh Thuan, Vietnam. There are eight time periods corresponding to the annual production planning problem over an eight-year period. Hence, the number of possible scenarios for planning dragon fruit production and distribution over an 8-year period is 6562 (3 raised to the power of 8). The variability in the selling price of the three different varieties of dragon fruit cultivated and consumed in Vietnam is considered:

white-flesh, red-skinned dragon fruit; red-flesh, red-skinned dragon fruit; and white-flesh, yellow-skinned dragon fruit. The scenario tree for each selling price is assessed separately and subsequently merged into a unified tree, as shown in Figure 12.

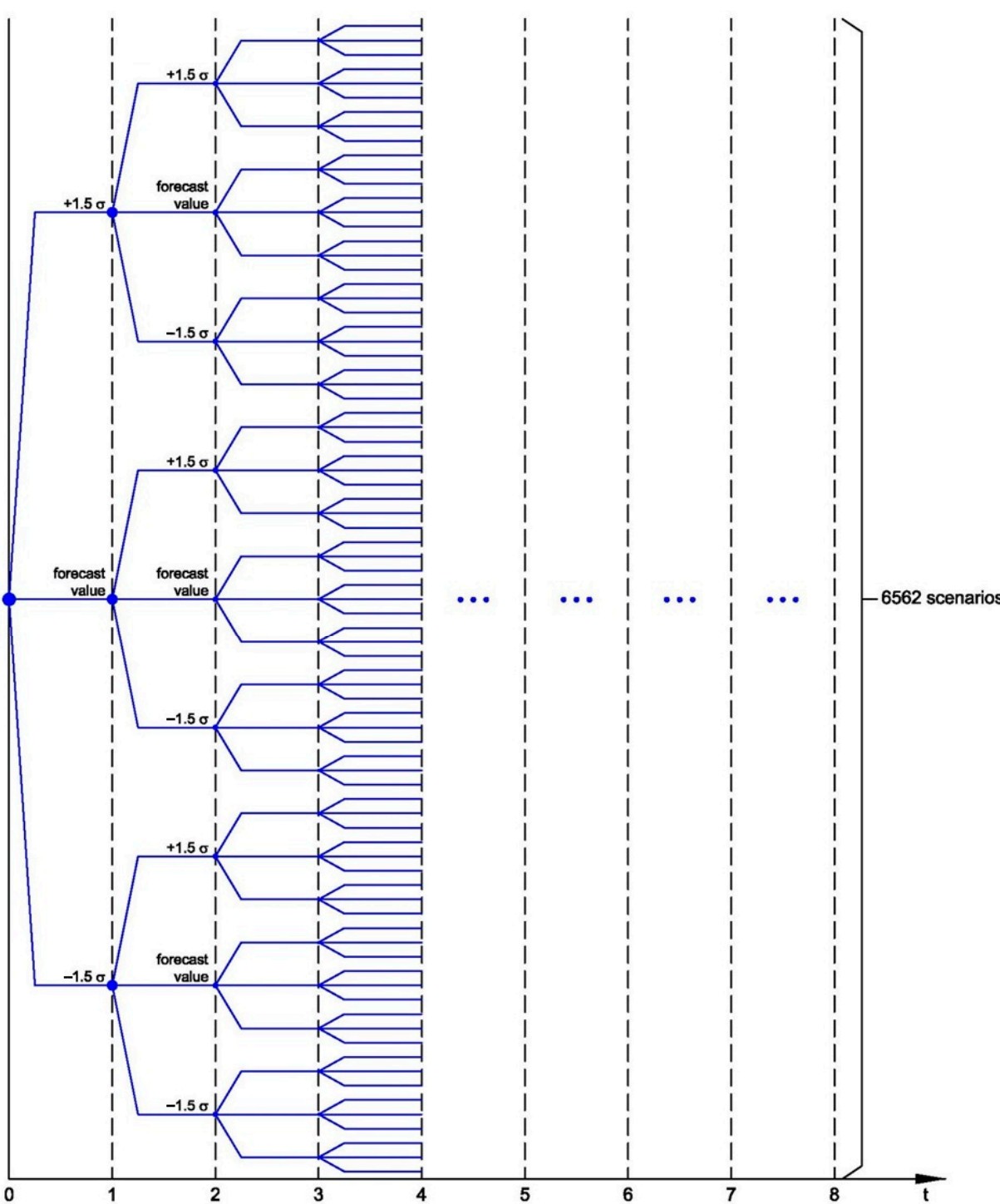

**Figure 12.** Scenario tree for selling prices of dragon fruits in an 8-year period.

In order to proceed, it is necessary to ascertain the probability associated with each potential outcome of the scenario tree by utilizing the Distribution Matching Problem (DMP) model that was presented in the preceding section. The statistical parameters utilized in the DMP model, including the mean value, standard deviation, variance, and covariance, are derived from the ARIMA forecasting model introduced in the preceding

stage. The likelihood value was calculated promptly upon acquiring the forecast value, upper value, and lower value. Hence, a total of 3280 iterations of the DMP model were conducted to determine the probability of all possible outcomes in the scenario tree over the course of the 8-year study period. An assumption was made that the chance of the selling price being the same for each outcome is equal for all three types of dragon fruit.

The stochastic programming model was implemented and solved using the Python programming language, resulting in a value of USD 34,574,662.31 after a computation time of around 9400 s.

### 4.2. Sample Average Approximation

The sample average approximation (SAA) is a method for solving stochastic optimization issues by utilizing Monte Carlo simulation. Given the extensive research period and the large number of scenarios involved in determining the optimal model for the production and distribution of fresh dragon fruit in Vietnam, the Monte Carlo sampling method is a prudent choice. This method is both simple and reliable, as the results obtained from the sample average approximation (SAA) are addressed using deterministic optimization techniques.

A Normal distribution was found to be followed by the selling price of dragon fruit, as can be seen from the historical data that was collected. The variable that represents the selling price of three different kinds of dragon fruit should be created and given random values in accordance with the Normal distribution. This objective function of the model should be computed using these random values, and the results should be recorded. The random values of the variables should be regenerated and reassigned. Perform another calculation of the objective function. Repeat the processes that were just described many times and then determine the average. The iterations of 200, 500, and 1000 are the ones that are suggested in this study. The results of the model are presented in Table 2.

**Table 2.** Results of SAA model.

| Iterations | Results of SAA Model |
|:---:|:---:|
| 200 | $44,941,772.13 |
| 500 | $44,777,497.37 |
| 1000 | $44,809,055.74 |

### 4.3. Chance-Constrained Programming

The objective function of the TSSP model that is introduced in the previous section has been rewritten in order to optimize the production and distribution of dragon fruit in Vietnam. Additionally, probabilistic constraints are added to specify the minimum probability with which the solution should satisfy the original constraints. In this study, the production and distribution plan is required to guarantee at least a percentage of chance of meeting customer demand (traders and wholesalers) in the 8-year period. The problem is solved in a sequential manner, with the anticipated levels of reliability being 80%, 85%, 90%, and 95% respectively. The outcomes of this procedure can be seen in the Figure 13.

The most optimal outcomes derived from those mentioned methodologies or approaches are briefly presented in Table 3.

Based on the results shown in Table 3, it is evident that the stochastic programming model outperforms the linear programming deterministic model. Specifically, the sample average approximation approach delivers more optimum outcomes compared to the other two stochastic methods. The chance-constrained programming technique is favored by us due to its ability to maximize profits while ensuring that 90% of client demands are addressed via the implementation of a powerful and robust SAA method.

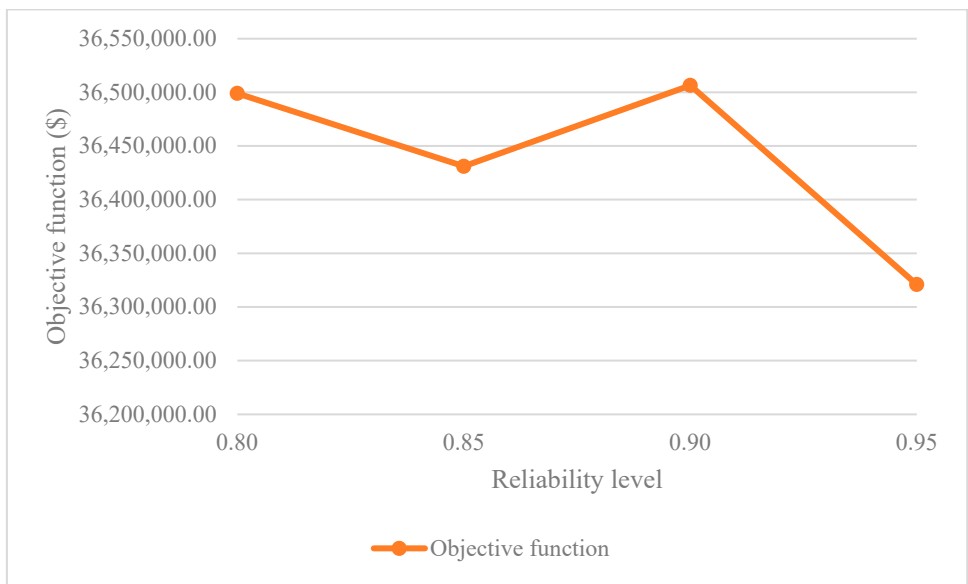

**Figure 13.** Objective function results of chance-constrained programming with different reliability levels.

**Table 3.** Summary of results.

|  | Deterministic Model (Nguyen et al., 2020) [13] | Scenario Tree Generation | Sample Average Approximation (200 Iterations) | Chance-Constrained Programming (*Rel* = 0.9) |
| --- | --- | --- | --- | --- |
| Objective function (CAD) | $16,815,925.82 | $34,574,662.31 | $44,941,772.13 | $36,505,446 |
| Number of variables |  | 3,307,250 | 126,632 | 101,306 |
| Number of constraints |  | 1,889,864 | 72,370 | 105,896 |
| Running time |  | 9472 s | 17 s | 6 s |

## 5. Conclusions

This work introduces a stochastic model for planning and organizing the production and distribution of dragon fruit in Vietnam, taking into account the unpredictable fluctuations in selling prices. Two methods, scenario tree generation and sample approximation average, are used to address the uncertainty of the issue by estimating the expected value of the objective function. Furthermore, a very effective and adaptable optimization approach called chance-constrained programming is suggested to consider the uncertainty in the dragon fruit trading price and its impact on meeting consumer demand.

Our stochastic approach in this paper encompasses the benefits of the linear programming model proposed by Nguyen et al. [13], which involves deciding whether to cultivate dragon fruit when its selling price is high or rising and discarding dragon fruits with lower prices or that are old. This stochastic model can additionally address the limitation of the deterministic model by effectively handling the unpredictability and ambiguity associated with dragon fruit selling prices. This helps dragon fruit producers and managers in gaining a more comprehensive understanding when making choices on the selection of dragon fruit types and cultivation areas in medium- and long-term plans.

Nevertheless, this paper does possess some constraints. Our research does not consider random elements such as demand and yield due to the challenges associated with data collection. Furthermore, the forecasting technique used for the stochastic approach is simplistic and lacks a high level of reliability.

Investigating the production and distribution of dragon fruit, as well as other fresh fruits, remains arduous and challenging. In the future, reliable sample and forecasting

techniques will be used to enhance the accuracy of planning. In addition, other stochastic methodologies are being investigated and implemented to provide decision-makers with more valuable information or a wider range of possibilities to consider when formulating long-term strategies.

**Author Contributions:** Conceptualization: T.-D.N., U.V. and T.N.-Q.; Data curation: T.-D.N. and T.N.-Q.; Format analysis: T.-D.N., U.V. and T.N.-Q.; Funding acquisition: T.N.-Q.; Investigation: T.-D.N. and U.V.; Methodology: T.-D.N., U.V. and T.N.-Q.; Software: T.-D.N., D.-H.P., H.-T.P., L.-K.P. and P.-C.N.; Supervision: T.N.-Q. and U.V.; Validation: T.-D.N., U.V., T.N.-Q., C.D. and M.A.; Visualization: C.D. and M.A.; Writing—original draft: T.-D.N., U.V. and T.N.-Q.; Writing—review and editing: all authors. All authors have read and agreed to the published version of the manuscript.

**Funding:** This research received no external funding.

**Institutional Review Board Statement:** Not applicable.

**Informed Consent Statement:** Not applicable.

**Data Availability Statement:** The data presented in this research are available on request from the corresponding author.

**Acknowledgments:** All authors express their gratitude to Vietnamese colleagues, collaborators, and farmers from Binh Thuan Province (Vietnam) for their data collection. We acknowledge the 911 research scholarship program from Vietnam International Education Development (VIED) addressed to T.-D.N.

**Conflicts of Interest:** The authors declare no conflicts of interest.

## Nomenclature

| | |
|---|---|
| ADB | Asian Development Bank |
| ARIMA | Autoregressive Integrated Moving Average |
| CCP | Chance-constrained programming |
| COVID-19 | Coronavirus disease 2019 |
| DF | Dragon fruit |
| DMP | Distribution Matching Problem |
| ECDF | Empirical Cumulative Distribution Function |
| ENSO | El Niño–Southern Oscillation |
| FAO | Food and Agriculture Organization |
| FFSC | Fresh fruit supply chain |
| KKT | Karush–Kuhn–Tucker |
| MILP | Mixed-integer linear programming |
| MONRE | Ministry of Natural Resources and Environment, Vietnam |
| MOSP | Multi-objective Stochastic Programming |
| PMP | Positive Mathematical Planning |
| RO | Robust optimization |
| SAA | Sample average approximation |
| SLP | Stochastic linear programming |
| SOFRI | Southern Horticultural Research Institute, Vietnam |
| SMILP | Stochastic mixed-integer linear programming |
| TSSP | Two-stage stochastic programming |

## Appendix A. Definition of Indices, Variables, and Parameters of the Stochastic Model

**Table A1.** Indices.

| Symbol | Description | Symbol for Max Value |
|--------|-------------|----------------------|
| $j$ | Dragon fruit species | $J$ |
| $k$ | Age classes | $K$ |
| $i$ | Traders | $I$ |
| $m$ | Wholesale markets | $M$ |
| $t$ | Time periods | $T$ |
| $s$ | Scenarios | $S$ |

**Table A2.** Variables.

| Indexer | Symbol | Description |
|---------|--------|-------------|
| year, crop, trader, scenario | $ST_{jits}$ | Amount of crop $j$ delivered to trader $i$ in time $t$ for scenario $s$ |
| year, crop, trader, scenario | $\epsilon^1_{jits}$ | Amount of crop $j$ under delivered to trader $i$ in time $t$ for scenario $s$ |
| year, crop, market, scenario | $SWM_{jmts}$ | Amount of crop $j$ delivered to wholesaler $m$ in time $t$ for scenario $s$ |
| year, crop, market, scenario | $\epsilon^2_{jmts}$ | Amount of crop $j$ under delivered to wholesaler $m$ in time $t$ for scenario $s$ |
| year, crop, scenario | $SB_{jts}$ | Amount of crop $j$ harvested for byproducts in time $t$ for scenario $s$ |
| year, crop, age | $X_{jkt}$ | Area of crop $j$ planted in time $t$ within age class $k$ |
| year, crop, age | $Z_{jkt}$ | Area cut of crop $j$ of age class $k$ in time $t$ |
| year, crop | $Y_{jt}$ | Area newly cultivated with crop $j$ in year $t$ |
| year | $F_t$ | Quantity of permanent employees in $t$ |
| year | $Hire_t$ | Part-time employees recruited during time $t$ |

**Table A3.** Parameters.

| Indexer | Symbol | Description |
|---------|--------|-------------|
| | $L$ | Amount of land available |
| | | Maximum lighting per hectare |
| | | Maximum water per hectare |
| age | $w_k$ | Water required per hectare for age class $k$ |
| age | $cwater$ | Cost of required water per hectare |
| crop | $v_j$ | Lighting required per hectare for crop $j$ |
| crop | $clighting$ | Cost of required light per hectare |
| year | $cr_t$ | Cost of cutting per hectare during time $t$ |
| year | $ch_t$ | Cost of harvesting per hectare during time $t$ |
| year | $cbp_t$ | Processing costs per ton during time $t$ |
| year | $clabf_t$ | Periodic cost of fixed staff |
| year | $clabp_t$ | Cost of labor for part-time employees each period |
| year | $R_t$ | Number of employees required to cut one hectare |
| year | $H_t$ | Number of employees required to harvest one hectare |
| year | $P_t$ | Number of employees required to plant one hectare |
| crop, age | $I_{jk}$ | Initial area of crop $j$ of age class $k$ |
| year, crop | $f_{jt}$ | Demand for byproducts of crop $j$ in period $t$ |
| year, crop | $u_{jt}$ | Minimum planting area per crop $j$ in period $t$ |
| year, crop | $cp_{jt}$ | Cost per hectare of planting for crop $j$ in period $t$ |

**Table A3.** *Cont.*

| Indexer | Symbol | Description |
|---|---|---|
| year, scenario | $\rho_s$ | Estimated price probability of scenario $s$ |
| year, crop, age | $y_{jkt}$ | Production in tons per hectare of crop $j$ in age class $k$ during the given period $t$ |
| year, crop, market | $e_{jmt}$ | The wholesale market's demand for crop $j$ during that time $t$ |
| year, crop, market | $cPNT2_{jmt}$ | Penalty for wholesaler $m$ not satisfying demand for each ton of crop $j$ during period $t$ |
| year, crop, scenario | $r_{jts}$ | Price per ton of byproducts in period $t$ in scenario $s$ |
| year, crop, trader | $d_{jit}$ | The trader's demand for crop $j$ during that time $t$ |
| year, crop, trader | $cPNT1_{jit}$ | Penalty for trader $i$ not satisfying demand for each ton of crop $j$ during period $t$ |
| year, crop, market, scenario | $p_{jits}$ | For trader $i$ in period $t$ in scenario $s$, the price per ton of crop $j$ |
| year, crop, trader, scenario | $q_{jmts}$ | For wholesaler $m$ in period $t$ in scenario $s$, the price per ton of crop $j$ |

## Appendix B. The Mathematical Model of Distribution Matching Problem

Indices:

Different species of crop $\qquad j$

The outcomes (branches) from the root node $\qquad o$

Variables:

Probabilities of outcomes $o$ $\qquad prob_o$

Positive and negative variances of crop $j$ calculated from the tree $\qquad var_j^+, var_j^-$

Positive and negative co-variances of crop $j$ and $j'$ calculated from the tree $\qquad cov_{jj'}^+, cov_{jj'}^-$

Positive and negative deviations of crop $j$ calculated from ECDF $\qquad \delta_{jo}^+, \delta_{jo}^-$

Parameters:

Uncertain parameters of the SP problem (prices) $\qquad x_{jo}$

Mean values of price dataset of crop $j$ $\qquad Mean_j$

Variance values of price dataset of crop $j$ $\qquad Var_j$

Co-variance values of price dataset between crop $j$ and crop $j'$ $\qquad CoV_{jj'}$

Standard deviation value of crop $j$ $\qquad Std_j$

Number of observations of crop $j$ dataset $\qquad n_j$

Weight value of variance of crop $j$ price $\qquad w\_var_j$

Weight value of co-variance crop $j$ and crop $j'$ $\qquad w_{cov jj'}$

Weight value of deviations of crop $j$ $\qquad w_{jo}$

Objective

$$\min Z = \mu + \gamma + \xi$$

S.t.

$$\sum_{o=1}^{O} prob_o = 1$$

$$\sum_{o=1}^{O} \left( x_{jo} \times prob_o \right) = Mean_j$$

$$\sum_{o=1}^{O} \left( x_{jo} - Mean_j \right)^2 prob_o + var_j^+ - var_j^- = Var_j$$

$$\sum_{o=1}^{O} \left( x_{jo} - Mean_j \right) \left( x_{j'o} - Mean_{j'} \right) prob_o + cov_{jj'}^+ - cov_{jj'}^- = CoV_{jj'}$$

$$\hat{ECDF}(x_{jo}) - \sum_{o'=1}^{o} prob_{o'} = \delta_{jo}^{+} - \delta_{jo}^{-} \quad o = 1\dots O$$

With

$$\hat{ECDF}(x_{jo}) = \Phi\left[\frac{x_{jo} - Mean_j}{\sqrt{Var_j}}\right]$$

$$\mu \geq w\_var_j \times v_j^{+}$$

$$\mu \geq w\_var_j \times v_j^{-}$$

$$\gamma \geq w\_cov_{jj'} \times cov_{jj'}^{+}$$

$$\gamma \geq w\_cov_{jj'} \times cov_{jj'}^{-}$$

$$\xi \geq \omega_{jo} \times \delta_{jo}^{+}$$

$$\xi \geq \omega_{jo} \times \delta_{jo}^{-}$$

$$var_j^{+}, var_j^{-}, cov_{jj'}^{+}, cov_{jj}^{-}, \delta_{jo}^{+}, \delta_{jo}^{-} \geq 0$$

$$prob_o \in [0,1]$$

$$x_{jo} \leq x_{j,o+1}$$

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
