# Peer review of "Stochastic Modelling Frameworks for Dragon Fruit Supply Chains in Vietnam under Uncertain Factors"

_sustainability, doi:10.3390/su16062423_

Round 1

Reviewer 1 Report

Comments and Suggestions for Authors

The article is well-constructed, written on a relevant topic, and contains interesting scientifically based results. To increase the scientific value of the article, the authors could make certain improvements. In particular, in the section Results and discussion, it is advisable to pay more attention to the comparison of the results obtained by the authors with the work of other scientists in this direction, to determine the differences that exist when applying different methodologies. This will make it possible to more thoroughly formulate results limitations and improve the logic of the presentation of the material.

Reviewer 2 Report

Comments and Suggestions for Authors

Dear authors,

Thank you for submitting your manuscript to the journal. I would like to offer you some advice.

1.     The abstract fails to succinctly summarize the entire document and does not clearly articulate the ultimate conclusions of the article and the advantages of the research findings. Additionally, abstracts typically do not include references.

2.     In section "2.2. Emerging trends of modelling frameworks for uncertainty and climate effects", the explanations of the methodologies are cumbersome and streamlining is recommended.

3.     When modeling, the description related to the symbols used is placed in a plan that is not conducive to cross-referencing, can it be changed in a better way?

4.     The article only provides different varieties of dragon fruit sales price data, it is recommended to add different varieties of dragon fruit in different regions of the sales volume and other related information.

5.     Incorrect numbering of charts, as in figures 8 and 9(line 787,line 789).

6.     Lines 858~863, i.e., "Based on the results... robust SAA method" part, according to the conclusion part of Table 3, when the three methods are compared, the results obtained by chance-constrained planning relative to the sample average approximation are more different, and the reasons for choosing the chance-constrained planning technique are not sufficient.

7.     The overall writing of the article lacks smoothness, making it read somewhat awkwardly. It appears somewhat cumbersome in parts, and there are instances, such as lines 244-248, where logical errors are evident. It is recommended to undergo revisions and optimizations for improvement.

Reviewer 3 Report

Comments and Suggestions for Authors

The work presents scientific wealth, but it is suggested:

  • Improve the summary, it does not include bibliographic references.
  • The contribution of the work is unclear.
  • The introduction should include: Background, Contributions and novelty of the research,
  • Scientific issues and research objective.
  • Review bibliographic references, ensuring that there are no self-citations.
  • Include sections referring to future research, implications and limitations of the job.

Reviewer 4 Report

Comments and Suggestions for Authors

The purpose of this article is to study

the uncertainty factors associated with the production and distribution of fresh fruits in Vietnam.

The article takes into account all the main factors affecting the production and distribution of dragon fruit, which is an advantage of research. However, political factors that also affect the cost and volume of fruit sales are not considered.

The manuscript is written in understandable language and is well structured. The citations are relevant and correct. The presented work is scientifically justified, the methodology is suitable for supply chain research in conditions of uncertain factors.

The presented drawings sufficiently reflect the essence of the research, are easy to read and interpret.

The presented conclusions reflect the results of the research.
